# Determinants of practice location choices among physicians and medical students in Mali: Insights into addressing medical deserts through evidence-based strategies

Issa Kalossi [1]*, Dielika Coulibaly[2], Kassoum Alou N'Diaye[1], Modibo Salia Drame[3], Djibril Sissoko [4], Thiery Almont[5], Kassoum Kayentao[1]

**1** Faculté de Médecine et d'Odontostomatologie, Université des Sciences des Techniques et des Technologies de Bamako, Bamako, Mali, **2** Direction régionale de la santé de Mopti, Centre de santé de référence de Djenne, Djenne, Mali, **3** National Institute of Nutrition and Food Technology (INNTA), Nutrition Surveillance and Epidemiology in Tunisia (SURVEN), Tunis, Tunisia, **4** Institut National de Prévoyance Sociale (INPS), Bamako, Mali, **5** Centre Hospitalo-Universitaire de Martinique, Pôle Cancérologie Hématologie Urologie, Fort-de-France, Martinique

* issakalossi@gmail.com

## Abstract

The shortage of medical professionals in rural and remote areas is a global issue that significantly challenges equitable healthcare delivery. Worldwide, various studies have examined the motivations of medical professionals in choosing their practice location. However, for Mali, this topic remains underexplored, motivating us to conduct this study to identify factors influencing physicians' workplace decisions in Mali. We conducted a cross-sectional study targeting physicians and final-year medical students. Using simple random sampling, we selected 358 respondents, 69% of whom were physicians. Data were collected via structured questionnaires assessing sociodemographic characteristics, professional aspirations, and factors influencing workplace preferences. Multivariate logistic regression was used to identify factors independently associated with rural practice preference. Approximately 38% of the respondents preferred rural areas for their practice, primarily citing career development and opportunities for continuing education (38%). The likelihood of choosing rural practice was higher (OR = 5.09; 95% CI = 2.52-10.8) among participants with family residing in rural areas. This study highlights both personal and systemic factors shaping physicians' workplace choices in Mali. Interventions aimed at strengthening rural practice appeal should prioritize professional development opportunities and support systems for those with rural backgrounds. Tailored strategies addressing these motivations could enhance equitable healthcare workforce distribution in Mali.

**Data availability statement:** All relevant data are within the paper and its Supporting Information files.

**Funding:** The author(s) received no specific funding for this work.

**Competing interests:** The authors have declared that no competing interests exist.

## Introduction

A medical desert is characterized by a shortage of medical professionals in a specific geographical area, which represents a global concern that poses significant challenges to equitable healthcare delivery. Both developed and developing countries report an uneven geographical distribution of medical professionals, favoring urban and affluent areas, despite the fact that rural communities experience greater health problems [1,2]. In 2012, more than half of the world's population was living in rural areas, whereas less than a quarter of health professionals worked there [3]. This spatial disparity in the distribution of healthcare workers means that those who most need healthcare services receive the poorest care, reflecting Hart's "Inverse Care Law" [4].

In Mali, community health centers (CSComs) represent the primary point of contact between the population and medical personnel. Nationally, only 32% of community health centers (CSCOMs) are staffed with a physician. This proportion reaches 98% in Bamako, but drops to just 29% in CSCOMs located outside the capital, highlighting a stark urban–rural disparity in physician distribution [5], this is well below the standards set by the World Health Organization [6]. Additionally, the National Institute of Statistics estimates, in its latest report (2018), an average of 0.86 physicians per 10,000 inhabitants in rural regions versus 3.73 physicians per 10,000 inhabitants in Bamako [7]. Several factors contribute to this imbalance. Mali's healthcare infrastructure is highly centralized, with most specialized services, advanced diagnostics, and medical training institutions located in Bamako. Medical education itself is exclusively urban-based, and opportunities for rural clinical exposure or training remain limited. As a result, medical students and early-career doctors often have little experience with rural health systems and may be less inclined to consider long-term practice in such environments.

Despite the critical importance of health workforce distribution, to the best of our knowledge, no large-scale study has been conducted in Mali to gather medical professionals' opinions on factors influencing their choice of practice location. Furthermore, this topic remains poorly documented in Mali's unique context. This gap in the literature limits the ability to design context-specific strategies to address rural health workforce shortages. This motivated us to investigate the main factors influencing the choice of practice location among physicians and final-year medical students in 2023. The findings of this study can inform targeted measures, public health actions, and workforce planning.

## Methods

### Ethics statement

This study received ethical approval from the National Ethics Committee for Health and Life Sciences (CNESS) in Mali, under reference number 23/04/MSDS/CNESS. To ensure the protection of respondents' personal data, we used an anonymous questionnaire. Each participant was assigned an auto-generated number by the software in response order, preventing individual identification. As paper-written consent

was not feasible for an online survey, respondents were informed about the objectives of the study and their voluntary participation. By completing and submitting the questionnaire, respondents were considered to have provided their informed consent to participate in the study.

## Study design

We conducted a cross-sectional study in Mali, surveying physicians and final-year medical students. Data were collected from May 1st to July 30th, 2023.

## Study population

Our study population included practicing physicians registered with the national medical council of Mali and final-year medical students. Unregistered physicians and students who were not in their thesis year were excluded.

## Sampling method and sample size determination

We employed simple random sampling (without replacement) from the list of registered physicians and final-year medical students. To determine the minimum sample size (n), we used the following formula: [8].

$$n = z^2 * \frac{p(1-p)}{m^2}.$$

n = sample size,
z = confidence level (1.96)
p: estimated proportion of the population with the choice of rural area.
m: margin of error
This calculation yielded a minimum required sample size of 347. Accounting 2% of the sample size as refusal rate, the final expected sample size was 354 participants.

## Data collection tools and procedures

Data were collected using an ad hoc questionnaire, tested prior to the study with 10 students and 10 physicians. These participants were not included in the final sample to avoid any bias in the results. After this pilot test, minor changes were made to questionnaire to ensure clarity, relevance, and improved understanding of the questions by participants. It was inspired by prior studies on similar topics [9–17] and included cafeteria-style response options addressing the most frequently cited reasons during interviews. The questionnaire also captured sociodemographic and professional factors influencing health professionals' practice location preferences. Designed in Microsoft Forms, all key variables were mandatory to minimize missing data. The form link was distributed to selected participants via Short Message Service (SMS).

## Statistical analysis

The choice of practice location (urban or rural) served as the primary dependent variable, while other factors were treated as independent variables. A descriptive analysis was performed, with results presented in frequency tables. Statistical tests, including Pearson's Chi-squared test and Fisher's exact test, were used to compare proportions. To estimate odds ratios (ORs) and 95% confidence intervals (CIs) for each factor, univariate and multivariate logistic regression models were employed using all the key variables. Non-significant variables were removed from the final model using a stepwise approach based on the Akaike Information Criterion (AIC). A p-value ≤ 0.05 was considered statistically significant. Data analysis was performed using R software version 4.

## Results

At the end of the study, we collected 358 responses, with a response rate of 94.2%. The average age of the study population was 31 years (physicians = 34; students = 26), with a standard deviation of 6 years. For physicians, the average professional experience was 5 years, ranging from 0 to 32 years. Male respondents were predominant, accounting for 83.80% of participants, resulting in a male-to-female ratio of 5.17. Approximately 64.25% of respondents had urban family residences. Among the 358 respondents, 191 (53.35%) were born and/or raised in urban areas.

Physicians constituted 68.72% (246/358) of the study population, among the 246 physicians surveyed, 76.42% reported working in the private sector. Bamako was the practice location for 51.22% of respondents, while the remaining 48.78% worked in other regions. About 69.27% of respondents reported no prior experience working or interning in rural settings (Table 1).

Career development opportunities and access to continuing education were cited as the primary reasons for choosing a practice location by the majority of respondents (37.71%). This was followed by factors such as family or personal factors. Financial reasons were the primary motivator for only 6.42% of respondents (Table 2).

(Table 3) When asked about their choice of practice location, less than 38% (135/358) of respondents indicated a preference for working in rural areas if given the choice. The intention to work in rural areas was higher among male respondents compared to female respondents, at 39% versus 31%, respectively; however, this difference was not statistically significant according to the Chi-squared test. Respondents with family residing in rural areas were significantly more likely to express a preference for rural practice compared to those without rural family ties, at 57.81% versus 26.52%. This difference was statistically significant ($p < 0.001$).

**Table 1. Sociodemographic and professional characteristics.**

| Characteristics | Count | Percentage (%) |
|---|---|---|
| **Age [Mean ± Sd]** | | [31 ± 6] |
| **Year of professional experience: Mean [Range]** | | 5 [0-32] |
| **Sex** | | |
| Male | 300 | 83.80 |
| Female | 58 | 16.20 |
| **Family residence** | | |
| Urban | 230 | 64.25 |
| Rural | 128 | 35.75 |
| **Rural Origins** | | |
| Yes | 167 | 46.65 |
| No | 191 | 53.35 |
| **Status** | | |
| Physician | 246 | 68.72 |
| Student | 112 | 31.28 |
| **Sector of employment** | | |
| Public sector | 58 | 23.58 |
| Private sector | 188 | 76.42 |
| **Practice location** | | |
| Bamako (capital city) | 126 | 51.22 |
| Other regions | 120 | 48.78 |
| **Rural experience** | | |
| Yes | 110 | 30.73 |
| No | 248 | 69.27 |

**Table 2. Distribution by motivational factors for choosing a practice location.**

| Motivational Factors | Reasons | |
|---|---|---|
| | Primary Reason n (%) | Secondary Reason n (%) |
| Career development/ Continuing education | 135 (37.71) | 81 (24.40) |
| Family/Personal factors | 99 (27.65) | 92 (27.71) |
| Technical facilities/Infrastructure | 51 (14.25) | 92 (27.71) |
| Accountability | 32 (8.94) | 27 (8.13) |
| Financial reasons | 23 (6.42) | 40 (12.05) |
| Other reasons | 18 (5.03) | – |

**Table 3. Personal and family factors associated with the choice of practice location.**

| Factors | Intention to Work in Rural Areas n (%) | p value | Univariate Analysis | | | Multivariate Analysis | | |
|---|---|---|---|---|---|---|---|---|
| | | | OR | 95% CI | p value | OR | 95% CI | p value |
| **Gender** | | | | | | | | |
| Male | 117 (39.00) | 0,3 | 1.00 | — | | 1.00 | — | |
| Female | 18 (31.03) | | 0.70 | 0.38 – 1.27 | 0.3 | 0.67 | 0.30 – 1.47 | 0.3 |
| **Family residence** | | | | | | | | |
| Urban | 61 (26.52) | <0,001 | 1.00 | — | | 1.00 | — | |
| Rural | 74 (57.81) | | 3.80 | 2.41 – 6.03 | <0.001 | 5.09 | 2.52 – 10.8 | <0.001 |
| **Rural origins** | | | | | | | | |
| No | 44 (26.35) | <0,001 | 1.00 | — | | 1.00 | — | |
| Yes | 91 (47.64) | | 2.54 | 1.64 – 4.00 | <0.001 | 0.87 | 0.41 – 1.78 | 0.7 |
| **Status** | | | | | | | | |
| Physician | 84 (34.15) | 0,039 | 1.00 | — | | 1.00 | — | |
| Student | 51 (45.54) | | 1.61 | 1.02 – 2.54 | 0.04 | 2.05 | 1.14 – 3.72 | 0.017 |
| **Rural experience** | | | | | | | | |
| No | 107 (43.15) | 0,001 | 1.00 | — | | 1.00 | — | |
| Yes | 28 (25.45) | | 2.22 | 1.36 – 3.70 | 0.002 | 2.30 | 0.77 – 6.81 | 0.13 |

*Note: OR: Odds Ratio; IC, Confidence Interval.*

Similarly, rural practice intentions were statistically different for participants born or raised in rural areas compared to those who had only lived in urban settings. Participants with prior professional experience in rural settings, either through internships or work placements, were more likely to express an intention to work in rural areas compared to those without such experience (p < 0.001).

Regarding sociodemographic characteristics, neither univariate nor multivariate analyses showed a statistically significant association between gender and choice of practice location, with OR = 0.70 (95% CI = 0.38–1.27) and OR = 0.67 (95% CI = 0.30–1.47), respectively.

Respondents with family residing in rural areas were significantly more likely to choose rural practice in both univariate (OR = 3.80, 95% CI = 2,41 – 6,03) and multivariate (OR = 5.09; 95% CI = 2,52 – 10,8) analyses. Rural origin was significantly associated with the choice of practice location in univariate analysis (OR = 2.54; 95% CI = 1,64 – 4,00), but this factor was not significant in multivariate analysis (OR = 0.87; 95% CI = 0,41 – 1,78), which may suggest a confusing factor.

Having rural experience was a factor favoring the intention to work in rural areas, with OR = 2.22 (95% CI = 1.36–3.70) in univariate analysis and OR = 2.30 (95% CI = 0.77–6.81) in multivariate analysis.

Students were significantly more likely than practicing physicians to intend to work in rural areas, with an OR of 1.61 (95% CI = 1.02–2.54) in univariate analysis and 2.05 (95% CI = 1.14–3.72) in multivariate analysis.

Factors associated with the choice of practice location are summarized in Table 3. In multivariate analysis, motivations related to access to technical infrastructure (OR = 0.01; 95% CI = 0.00–0.04) and to career development or continuing education (OR = 0.01; 95% CI = 0.00–0.05) were significantly associated with a lower likelihood of choosing rural practice.

Conversely, accountability was strongly associated with rural practice in univariate analysis (OR = 31.6; 95% CI = 9.30–197), but this association was not significant in the multivariate model (OR = 0.69; 95% CI = 0.03–8.13).

Motivations based on personal or family reasons showed no significant association in univariate analysis (OR = 1.17; 95% CI = 0.73–1.88), but were negatively associated with rural practice in the multivariate model (OR = 0.02; 95% CI = 0.00–0.13).

Financial motivations were also not significant in multivariate analysis, though a negative association was observed (OR = 0.03; 95% CI = 0.00–0.22) (Table 4).

According to the final model (Fig 1), the family residence of the respondents had a significant influence on their choice of practice location. Students were also more likely to choose rural areas compared to practicing physicians. Having prior experience in rural settings was another factor positively influencing the choice of rural practice.

Conversely, financial motivation, personal or family reasons, career opportunities, continuing education, and access to technical facilities were all significantly and negatively associated with the decision to practice in rural areas.

## Recommendations and proposed solutions

To promote an equitable distribution of physicians across the country, our results have highlighted the following key approaches:

**Recruitment.** Terms such as "recruitment," "direct integration," or "massive recruitment" were frequently mentioned by respondents. In Mali, recruitment into the state public service is conducted through competitive examinations. Out of an

**Table 4. Professional factors influencing the choice of practice location.**

| Factors | Intention to Work in Rural Areas n (%) | p value | Univariate Analysis | | | Multivariate Analysis | | |
|---|---|---|---|---|---|---|---|---|
| | | | OR | IC à 95% | p value | OR | IC à 95% | p value |
| **Access to technical facilities/infrastructure** | | | | | | | | |
| No | 128 (41.69) | <0.001 | 1.00 | — | | 1.00 | — | |
| Yes | 7 (13.73) | | 0.22 | 0.09 – 0.48 | <0.001 | 0.01 | 0.00 – 0.04 | <0.001 |
| **Career development/continuing education** | | | | | | | | |
| No | 105 (47.09) | <0.001 | 1.00 | — | | 1.00 | — | |
| Yes | 30 (22.22) | | 0.32 | 0.20 – 0.52 | <0.001 | 0.01 | 0.00 – 0.05 | <0.001 |
| **Personal or family reasons** | | | | | | | | |
| No | 95 (36.68) | 0.5 | 1.00 | — | | 1.00 | — | |
| Yes | 40 (40.40) | | 1.17 | 0.73 – 1.88 | 0.5 | 0.02 | 0.00 – 0.13 | <0.001 |
| **Accountability** | | | | | | | | |
| No | 105 (32.21) | <0.001 | 1.00 | — | | 1.00 | — | |
| Yes | 30 (93.75) | | 31.6 | 9.30 – 197 | <0.001 | 0.69 | 0.03 – 8.13 | 0.8 |
| **Financial reasons** | | | | | | | | |
| No | 124 (37.01) | 0.3 | 1.00 | — | | 1.00 | — | |
| Yes | 11 (47.83) | | 1.56 | 0.66 – 3.66 | 0.3 | 0.03 | 0.00 – 0.22 | 0.003 |

*Note: OR: Odds Ratio; IC, Confidence Interval.*

PLOS Global Public Health

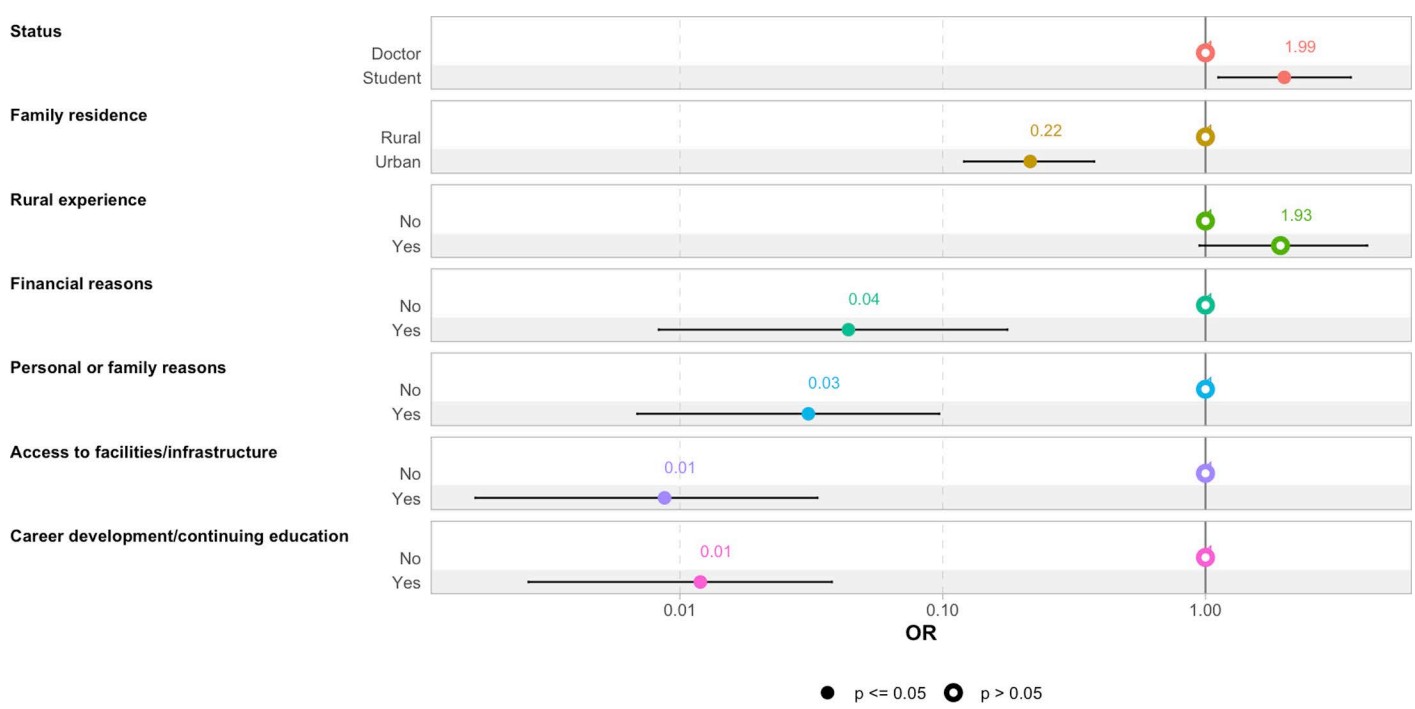

**Fig 1. Final multivariate regression model of associated factors.**

average of 350 physicians trained annually, only about twenty are integrated into the public service each year. This limited recruitment may contribute to the shortage of physicians in certain remote or underserved areas. In the absence of large-scale national recruitment, some respondents suggested community-based recruitment strategies, explicitly specifying the practice location, to address geographic disparities.

**Decentralization.** The decentralization of technical facilities and improved access to healthcare infrastructure were identified by some respondents as factors that could encourage physicians to settle in rural areas. The decentralization of continuing medical education was also proposed as an incentive. Currently, in Mali, access to general medical education and medical specialization is only available in Bamako (the capital city).

**Financial incentives and benefits.** Financial motivation through increased salaries, special allowances for physicians working in rural areas, and other incentive-based benefits were suggested by several respondents as potential solutions to reduce medical deserts.

**Political will/ leadership.** The lack of political commitment in the health sector was also highlighted. Some participants stated that strong political will is essential to address the issue of medical deserts effectively.

**Rural internships.** The establishment of a mandatory rural internship program for final-year medical students was proposed. This measure could enable early-career physicians to acquire foundational knowledge and practical skills related to rural medical practice.

## Discussion

To the best of our knowledge, this study, which aimed to analyze the factors associated with the choice of practice location among physicians and final-year medical students, is the first of its kind in our context. It provided insights into the motivations and influencing factors behind the practice location preferences of physicians and medical students.

## Family residence

The results revealed a strong association between family residence and the respondents' choice of practice location. Having family in rural areas positively influenced the decision to practice in rural settings. Among the 128 respondents whose families resided in rural areas, approximately 58% chose rural practice, a statistically significant difference. This factor was also associated with the choice of practice location in univariate (OR = 3.80, 95% CI = 2,41 – 6,03) and multivariate analyses (OR = 5.09, 95% CI = 2,52 – 10,8). This finding aligns with other studies conducted in Botswana [18], Ethiopia [19] and Nepal [20]. However, other research, such as that by Liu et al. [21], Playford et al. [22] and Grobler et al. [23] found family residence to be less influential on physicians choice of practice location. This may suggest differences between socio economic and cultural aspects in these countries.

## Rural experience

A statistically significant relationship was observed between rural experience and the choice of practice location in univariate analysis. Respondents with rural experience were 2.22 times (crude OR) more likely to intend to practice in rural areas than those without such experience. However, this difference was not significant in multivariate analysis (95% CI = 0.77–6.81). This result is consistent with findings from other countries [22,24,25].

## Status

Our multivariate logistic regression model also identified status as a significant factor influencing respondents' decisions. Among the 112 students in our sample, nearly 46% expressed an intention to practice in rural areas, compared to 34% of practicing physicians. This difference was statistically significant according to the Chi-squared test. At this stage, they may be more receptive to rural practice, especially if they perceive it as a meaningful opportunity for impact. In multivariate regression, students were 2.05 times more likely than physicians to intend to work in rural areas (95% CI = 1.14–3.72). This finding aligns with those of Raul et al. [26], who reported a strong rural practice intention among 21% of students.

## Financial motivation

Our study found that financial motivation was negatively associated with the decision to practice in rural areas. This change suggests that the initial positive associations were likely confounded by other variables. This shift may also reflect a suppression effect and highlight the complexity of decision-making processes. Physicians and students driven by financial incentives were more likely to prefer urban practice, with an OR of 0.03 (95% CI = 0.00–0.22) in multivariate analysis. This result is comparable to studies conducted in Kenya [27] and Ghana [28], where financial motivation was a key factor influencing respondents' choices.

## Personal or family reasons

Personal or family reasons were also associated with the respondents' choice of practice location in multivariate analysis. However, contrary to studies in Tanzania [29] and a 2016 systematic review [30], our findings revealed that participants citing personal or family reasons were more likely to prefer urban practice. This factor was statistically significant after adjustment (OR = 0.02; 95% CI = 0.00–0.13]).

## Access to infrastructure and technical facilities

Access to infrastructure and technical facilities also emerged as a motivational factor. Approximately 14% of respondents cited this as their primary reason for choosing a practice location, with nearly 28% mentioning it as a secondary reason. Over 86% of respondents motivated by this factor preferred urban practice. This preference was significant in both univariate (OR = 0.22) and multivariate analyses (OR = 0.01; 95% CI = 0.00–0.04). Our result was comparable to a study

conducted in Kenya [27] and Ghana [28], where inadequate access to electricity, equipment, and transport was found to be a determining factor in the choices made by the respondents. The lack of housing, inadequate remuneration for health staff, and the poor condition of healthcare facilities contribute to an unfavorable work environment. Inadequate working conditions, combined with low job satisfaction and stability, were cited as factors that could demotivate healthcare workers and impact their decision to practice in rural areas, according to the study by Dielemann et al. [12] and Willis-Schattuck et al. [31]. Other authors have also reported that housing is one of the factors that attract and retain rural physicians [32]. Hospital accommodation is often the only realistic option for rural physicians because many hospitals are in extremely underdeveloped areas, with inadequate housing available in communities outside the hospital.

## Career development and continuing education

Another significant motivation influencing the choice of practice location among respondents was the opportunity for career advancement and continuing education. In our study, 135 respondents (38%) stated that career progression and continuing education opportunities were the primary reasons for their decision, while 24% cited these as secondary reasons. Among participants whose primary reason was career development or continuing education, 78% opted for urban practice, a statistically significant difference based on the Chi-squared test ($p < 0.001$).

Respondents who prioritized career development or continuing education were more likely to choose urban settings as their practice location, as shown by univariate logistic regression (OR = 0.32; 95% CI = 0.20–0.52) and multivariate regression (adjusted OR = 0.01; 95% CI = 0.00–0.04). These findings align with those of a systematic review [31], which indicated that health professionals were reluctant to work in rural areas due to fewer career development opportunities compared to urban settings [33]. Studies have shown that health professionals feel proud and motivated when they perceive opportunities for progression.

## Accountability

Accountability was another factor described in the literature as significantly influencing the decision of health professionals to practice in rural areas. According to our study, accountability was cited as the primary reason by approximately 9% of respondents. Among these, nearly 94% expressed an intention to work in rural areas if given the choice, a statistically significant difference based on the Chi-squared test.

In logistic regression analysis, participants motivated by accountability were 32 times more likely (crude OR) to intend to work in rural areas compared to those citing other reasons. However, this difference was not statistically significant after adjustment for all other variables (OR = 0.69; 95% CI = 0.03–8.13). This shift may reflect the fact that these factors, while influential, are indirectly linked to practice choice and operate through more proximal motivational factors such as career opportunities or access to infrastructure. For example, a participant motivated by accountability may still be deterred from rural practice if technical support or development prospects are lacking. Some studies have similarly reported that feelings of belonging, and accountability are key determinants influencing participants' decisions to remain in rural areas [34].

## Rural origins

In our study, participants born or raised in rural areas accounted for approximately 47% of respondents. Among them, nearly 58% expressed an intention to practice in rural areas, a statistically significant difference based on the Chi-squared test ($p < 0.001$). Respondents with rural origins were 2.54 times more likely (crude OR) to choose rural practice compared to those with urban origins. However, this trend was not statistically significant after adjustment in multivariate logistic regression.

In contrast to our findings, other studies have reported statistically significant relationships between rural origin and the choice to practice in rural areas. These studies found that rural origin was positively associated with the preference for rural practice locations [22,24,25].

## Gender imbalance

We also evaluated the influence of other sociodemographic characteristics, such as gender, on respondents' choice of practice location. Female respondents were underrepresented in our study, accounting for only 16% of participants, which may be explained by the imbalance male/female ratio among physicians and medical student in Mali. A study conducted by the World Bank in seven African countries showed an underrepresentation of women among doctors and medical students, particularly in West Africa [35]. Among female respondents, 31% expressed an intention to work in rural areas compared to 39% of male respondents. However, this difference was not statistically significant (p = 0.3). In both univariate and multivariate analyses, female respondents were more inclined to choose urban settings. The ORs were 0.70 (95% CI = 0.38–1.27) and 0.67 (95% CI = 0.30–1.47), respectively. Similarly, findings from Sheppard et al. shown that women were less inclined to choose rural area [36].

## Strengths and limitations

One of the key strengths of this study lies in its originality, particularly the inclusion of final-year medical students as part of the study population. This group is rarely investigated in similar contexts, especially in Mali, despite representing the future workforce whose preferences could significantly shape the healthcare landscape in the coming years. By capturing the perspectives of both practicing physicians and students, our study offers a more comprehensive understanding of the determinants influencing practice location choices.

However, some limitations must be acknowledged. In fact, the expressed intentions of medical students regarding their future practice location may not necessarily translate into actual professional choices later in their careers. The existing literature suggests that career intentions often evolve under the influence of personal, professional, and contextual factors. Also, the geographic scope of the study may limit the generalizability of the findings to other regions beyond the surveyed areas. Finaly, we consider a risk of social desirability because respondents might give answers they believe are socially acceptable or valued.

These considerations reflect our commitment to a critical and reflexive interpretation of the results, ensuring that conclusions are drawn with appropriate caution.

## Conclusion

Medical deserts are a phenomenon affecting all countries worldwide, both developed and developing. This issue has serious consequences for public health, particularly in rural and remote populations. In Mali, this phenomenon is especially acute due to systemic limitations in infrastructure, centralized medical education, and insufficient rural deployment strategies. This study offers a clearer understanding of the personal, educational, and structural factors influencing the practice location choices of physicians and medical students. Key determinants such as family residence, rural exposure, and professional development opportunities were significantly associated with the intention to work in rural areas.

To address the shortage of healthcare providers in underserved regions of Mali, several concrete and context-specific recommendations emerge from our findings:

• Community-based recruitment and decentralized posting

• Incentivize rural practice

• Strengthen rural clinical exposure in medical training

• Decentralize continuing medical education

• Improve rural living and working conditions

• Promote stronger political commitment and leadership

These factors could serve as levers for ensuring the equitable distribution of physicians across the national territory.

## Author contributions

**Conceptualization:** Issa Kalossi, Thiery Almont, Kassoum Kayentao.

**Data curation:** Issa Kalossi.

**Formal analysis:** Issa Kalossi.

**Investigation:** Issa Kalossi.

**Methodology:** Issa Kalossi, Kassoum Alou N'Diaye, Thiery Almont, Kassoum Kayentao.

**Software:** Issa Kalossi.

**Validation:** Issa Kalossi.

**Visualization:** Issa Kalossi.

**Writing – original draft:** Issa Kalossi.

**Writing – review & editing:** Dielika COULIBALY, Kassoum Alou N'Diaye, Modibo Salia Dramé, Djibril Sissoko, Thiery Almont, Kassoum Kayentao.

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
