## [Decision Letter · Decision Letter 0]

PGPH-D-25-00341

Determinants of practice location choices among physicians and medical students in Mali: insights into addressing medical deserts through evidence-based strategies.

Dear Dr. Kalossi,

Thank you for submitting your manuscript to PLOS Global Public Health. After careful consideration, we feel that it has merit but does not fully meet PLOS Global Public Health’s publication criteria as it currently stands. Therefore, we invite you to submit a revised version of the manuscript that addresses the points raised during the review process.

We look forward to receiving your revised manuscript.

Kind regards,

Somayeh Hessam

Academic Editor

Journal Requirements:

1. We do not publish any copyright or trademark symbols that usually accompany proprietary names, eg (R), (C), or TM (e.g. next to drug or reagent names). Please remove all instances of trademark/copyright symbols throughout the text, including ® on page 2.

2. Please upload a copy of Figures 1 to 3 which you refer to in your text on page 7. Or, if the figure is no longer to be included as part of the submission please remove all reference to it within the text.

Additional Editor Comments (if provided):

Reviewers' comments:

Reviewer's Responses to Questions

**Comments to the Author**

1. Does this manuscript meet PLOS Global Public Health’s publication criteria?

Reviewer #1: Yes

Reviewer #2: Yes

Reviewer #3: Partly

Reviewer #4: Yes

Reviewer #5: Yes

2. Has the statistical analysis been performed appropriately and rigorously?

Reviewer #1: Yes

Reviewer #2: Yes

Reviewer #3: No

Reviewer #4: Yes

Reviewer #5: Yes

3. Have the authors made all data underlying the findings in their manuscript fully available (please refer to the Data Availability Statement at the start of the manuscript PDF file)?

Reviewer #1: Yes

Reviewer #2: Yes

Reviewer #3: Yes

Reviewer #4: Yes

Reviewer #5: Yes

4. Is the manuscript presented in an intelligible fashion and written in standard English?

Reviewer #1: Yes

Reviewer #2: Yes

Reviewer #3: Yes

Reviewer #4: Yes

Reviewer #5: Yes

Reviewer #1: Dear Author,

This is a well-structured manuscript tackling a critical issue in global health. Please take a look at my comments below.

Thank you.

Abstract:

The abstract effectively summarizes the study but could benefit from a clearer structure. To enhance readability, consider explicitly breaking it into Background, Methods, Results, and Conclusion. Instead of CI = [2.52-10.8], use 95% CI: 2.52–10.8 for standardization. Emphasize actionable takeaways, especially on how the findings can inform policy decisions.

Introduction:

The opening sentences could be condensed to avoid redundancy. The first two sentences make the same point. The citation [4] is useful, but a brief explanation of the principle would be helpful for readers unfamiliar with it. The data on CSComs and the urban-rural doctor ratio are strong points. However, make sure they align with the latest WHO recommendations for physician density.

Methods:

Well-described, but specify why final-year students were chosen rather than a broader range of students. The formula for sample size determination should be formatted properly. Kindly define z, p, and m.

Was the questionnaire validated for reliability (e.g., Cronbach’s alpha)? If your questionnaire included multiple questions assessing the same concept (e.g., factors influencing practice location), you should check whether responses to these questions were consistent using Cronbach’s Alpha.

Mention if there was a pilot test and any modifications made based on feedback. If you conducted a pilot test, mention how many people participated and what changes were made based on their feedback.

Result:

Ensure consistency in reporting statistical results. For instance, some p-values are explicitly stated (e.g., p < 0.001), while others are embedded within the text. Consider using a uniform format. Odds ratios (ORs) should always include confidence intervals (CIs) for clarity, which is well done in most instances. However, for the final model (Figure 3), it would be helpful to report specific ORs for each significant factor.

The average age and standard deviation are useful, but it would be beneficial to include a median age as well, considering the potential skewness of professional experience. Since gender distribution is highly imbalanced (83.80% male), a brief mention of whether this reflects the broader physician/student demographic in Mali would provide a better context.

The fact that rural origin was significant in univariate analysis but not in multivariate analysis suggests confounding variables. Consider briefly explaining why this might be the case. The term "accountability" as a motivation should be clearly defined. What specific aspects of accountability motivate respondents? Ethical obligations? Community responsibility?

Discussion:

The contradiction between Liu et al. (2020) and other studies regarding the role of family residence could be explained further. Does it reflect cultural differences, economic factors, or healthcare policies?

The study does not explicitly mention potential biases (e.g., self-selection bias, social desirability bias in responses). These should be acknowledged. Since 51.22% of respondents work in Bamako, the findings may not fully represent rural physicians. Consider discussing this limitation.

Minor Revision

Kindly go over the manuscript and make sure it is free of all minor grammatical errors. Kindly ensure all works cited are corrected and cited in the English language.

Excellent work on this study.

Reviewer #2: I would like to see some additional details in the discuss that pertain to the following:

1. Who bears the cost of medical education in the country and what is the contribution of students and doctors towards that cost? [the reason for this info is that very often when people pay for their education, they would want to recoup the costs from their pocket as soon as possible]

2. What proportion of doctors are in the private sector and what proportion in the public sector? Is the sample proportionate to these realities? Who are the major employers other than the government? What about those in the NGOs?

3. Any information or previous study that gives an understanding of who is preferred for medical consultations on the part of the community?

4. Finally any information on HRH analysis in terms of projections for the country by urban and rural populations.

Overall, the study is important and useful and adds to the body of knowledge from a country where very little is known about the healthcare professionals.

Reviewer #3: This paper describes the determinants of practice location choices among physicians and medical students in Mali. It should be publishable provided some extra work is done on the results and discussion section. The methods section and presentation of the results were not satisfactorily done and the interpretation of the results did not provide adequate depth. Additionally, there was no consistency in terminology with different terms being used interchangeably to refer to the same concept. There is a disparity where the authors use the term physician in the title and the term doctors throughout the study. The authors should rewrite their article to use either of the mentioned terms ie either physicians or doctors.

Reviewer #4: The authors have done a great job. Working on few areas would strengthen the manuscript further. Recommendations are as follows:

1. Expand the introduction to better connect Mali's healthcare challenges to the broader global rural healthcare crisis. Adding insights on Mali’s healthcare infrastructure, workforce distribution, and medical education system would provide necessary context.

2. The introduction should state the gap in literature more concisely and directly link this to the study objectives.

3. The references are heavily reliant on older studies. Incorporating more recent (post-2020) references would improve the manuscript’s relevance.

4. For methods, specifically authors must justify the chosen 2% refusal rate assumption in the sample size calculation.

5. Please provide a clear rationale for excluding non-thesis-year students and non-registered physicians.

6. Authros should clarify whether the questionnaire was pilot-tested. If so, describe what modifications were made based on pilot results.

7. Highly recommend incorporating visual aids such as bar charts, pie graphs, or infographics to present key findings more effectively.

8. While Odds Ratios (ORs) and Confidence Intervals (CIs) are reported, a brief interpretation of these results for non-statistical readers would improve clarity.

9. Since male respondents dominate the sample (83.8%), the potential impact of gender bias on findings should be explicitly discussed.

10. The discussion is overly descriptive. Include deeper insights into why career advancement deters rural practice and how this aligns with or diverges from global literature.

11. Mali’s unique social, economic, and healthcare dynamics need more emphasis to explain physician preferences.

12. Address the impact of the study’s gender imbalance and the differences in preference patterns between students and practicing physicians.

13. Expand the conclusion with practical, actionable recommendations tailored to Mali’s healthcare context..currently missing.

14. Strengthen the language in the conclusion to encourage policymakers to act on the findings.

15.. Emphasize concrete strategies such as rural internships, decentralizing education programs, or improving rural infrastructure.

Reviewer #5: REVIEW: Determinants of practice location choices among physicians and medical students in Mali: insights into addressing medical deserts through evidence-based strategies--Issa Kalossi, MD, MPH, et al

General:

This paper describes conducted a cross-sectional and prospective study targeting doctors and final-year medical students in Mali. The purpose of the study was to identify factors influencing doctors' workplace decisions and identify factors that may be relevant to addressing workforce shortages of medical doctors in this setting.

The reviewer appreciated the design of the survey, however the random section of 358 respondents was confusing. Why was the entire population of respondents not considered and why were specifically 358 responses randomly selected?

This reviewer is curious about why medical doctors were identified as the target population when many diverse professionals (nurses, laboratorians, therapists, etc.) make healthcare delivery possible. The narrowing of this definition of medical professionals could be described in the introduction and discussion.

This reviewer suggests sharing the survey in supplemental material.

Specific comments:

Results. It is customary in survey responses to provide the response rate (responses/total population surveyed). Kindly provide the response rate.

Tables: The asterisks signifying statistical significance were not reflected in the table, despite being defined in the key. This reviewer suggests updating the tables with these symbols.

Figure 1: The regression model contained French (oui/non) responses. For clarity, this reviewer suggests a consistent language throughout.

Discussion: The authors observe a gender imbalance in their response demographics. This should also be described as a limitation of findings between male and female respondents. For example, did the female candidates opt out of responding to the survey or does this represent the underlying distribution of male to female medical doctors? Could purposive sampling of the respondents to enrich female participants to better describe differences?

**Do you want your identity to be public for this peer review?** For information about this choice, including consent withdrawal, please see our Privacy Policy

Reviewer #1: No

Reviewer #2: **Yes: ** Lakshmi Narasimhan Balaji

Reviewer #3: **Yes: ** Sheillah Ansiima

Reviewer #4: No

Reviewer #5: **Yes: ** Linda S Barnes

---

## [Decision Letter · Decision Letter 1]

PGPH-D-25-00341R1

Determinants of practice location choices among physicians and medical students in Mali: insights into addressing medical deserts through evidence-based strategies.

Dear Dr. Kalossi,

Thank you for submitting your manuscript to PLOS Global Public Health. After careful consideration, we feel that it has merit but does not fully meet PLOS Global Public Health’s publication criteria as it currently stands. Therefore, we invite you to submit a revised version of the manuscript that addresses the points raised during the review process.

We look forward to receiving your revised manuscript.

Kind regards,

Somayeh Hessam

Academic Editor

Journal Requirements:

Additional Editor Comments (if provided):

Reviewers' comments:

Reviewer's Responses to Questions

**Comments to the Author**

Reviewer #1: All comments have been addressed

Reviewer #2: All comments have been addressed

Reviewer #3: (No Response)

Reviewer #5: All comments have been addressed

publication criteria?

Reviewer #1: Yes

Reviewer #2: Yes

Reviewer #3: Yes

Reviewer #5: Yes

3. Has the statistical analysis been performed appropriately and rigorously?

Reviewer #1: Yes

Reviewer #2: Yes

Reviewer #3: No

Reviewer #5: N/A

4. Have the authors made all data underlying the findings in their manuscript fully available (please refer to the Data Availability Statement at the start of the manuscript PDF file)?

Reviewer #1: Yes

Reviewer #2: Yes

Reviewer #3: Yes

Reviewer #5: Yes

5. Is the manuscript presented in an intelligible fashion and written in standard English?

Reviewer #1: Yes

Reviewer #2: Yes

Reviewer #3: Yes

Reviewer #5: Yes

Reviewer #1: Dear Author,

Thank you for thoroughly addressing all the concerns raised in the manuscript. Overall, the manuscript is now much stronger and aligns well with the journal’s standards.

Reviewer #2: As such the authors have responded to my comments and also of other reviewers adequately and i recommend its acceptance without any further changes

Reviewer #3: The author still needs to make major adjustments to the results section

Reviewer #5: Thank you for the opportunity to review.

**Do you want your identity to be public for this peer review?** For information about this choice, including consent withdrawal, please see our Privacy Policy

Reviewer #1: No

Reviewer #2: **Yes: ** Lakshmi Narasimhan Balaji

Reviewer #3: **Yes: ** Sheillah Ansiima

Reviewer #5: **Yes: ** Linda S Barnes

---

## [Decision Letter · Decision Letter 2]

Determinants of practice location choices among physicians and medical students in Mali: insights into addressing medical deserts through evidence-based strategies.

PGPH-D-25-00341R2

Dear Dr Kalossi,

We are pleased to inform you that your manuscript 'Determinants of practice location choices among physicians and medical students in Mali: insights into addressing medical deserts through evidence-based strategies.' has been provisionally accepted for publication in PLOS Global Public Health.

Best regards,

Somayeh Hessam

Academic Editor

Reviewer Comments (if any, and for reference):

Reviewer's Responses to Questions

**Comments to the Author**

Reviewer #1: All comments have been addressed

Reviewer #3: All comments have been addressed

Reviewer #5: All comments have been addressed

publication criteria?

Reviewer #1: Yes

Reviewer #3: Yes

Reviewer #5: Yes

3. Has the statistical analysis been performed appropriately and rigorously?

Reviewer #1: Yes

Reviewer #3: Yes

Reviewer #5: Yes

4. Have the authors made all data underlying the findings in their manuscript fully available (please refer to the Data Availability Statement at the start of the manuscript PDF file)?

Reviewer #1: Yes

Reviewer #3: Yes

Reviewer #5: Yes

5. Is the manuscript presented in an intelligible fashion and written in standard English?

Reviewer #1: Yes

Reviewer #3: Yes

Reviewer #5: Yes

Reviewer #1: All comment addressed.

Reviewer #3: Pg 13 the statement "Similarly, findins from Sheppard et al. shown that women were less inclined to chose rural area" correct to findings.

Reviewer #5: Thank you for your revisions.

**Do you want your identity to be public for this peer review?** For information about this choice, including consent withdrawal, please see our Privacy Policy

Reviewer #1: No

Reviewer #3: **Yes: ** Sheillah Ansiima

Reviewer #5: **Yes: ** Linda S Barnes
